# 4D Printing of Electroactive Triple-Shape Composites

**DOI:** 10.3390/polym15040832

**Published:** 2023-02-07

**Authors:** Muhammad Yasar Razzaq, Joamin Gonzalez-Gutierrez, Muhammad Farhan, Rohan Das, David Ruch, Stephan Westermann, Daniel F. Schmidt

**Affiliations:** 1Department of Materials Research and Technology, Luxembourg Institute of Science and Technology, ZAE Robert Steichen, L-4940 Hautcharage, Luxembourg; 2Institute of Active Polymers, Helmholtz-Zentrum Hereon, D-14513 Teltow, Germany; 3Department of Physics and Materials Science, University of Luxembourg, L-4365 Esch-sur-Alzette, Luxembourg

**Keywords:** 4D printing, additive manufacturing, triple-shape effect, electro-active composites, shape-memory polymers

## Abstract

Triple-shape polymers can memorize two independent shapes during a controlled recovery process. This work reports the 4D printing of electro-active triple-shape composites based on thermoplastic blends. Composite blends comprising polyester urethane (PEU), polylactic acid (PLA), and multiwall carbon nanotubes (MWCNTs) as conductive fillers were prepared by conventional melt processing methods. Morphological analysis of the composites revealed a phase separated morphology with aggregates of MWCNTs uniformly dispersed in the blend. Thermal analysis showed two different transition temperatures based on the melting point of the crystallizable switching domain of the PEU (*T*_m_~50 ± 1 °C) and the glass transition temperature of amorphous PLA (*T*_g_~61 ± 1 °C). The composites were suitable for 3D printing by fused filament fabrication (FFF). 3D models based on single or multiple materials were printed to demonstrate and quantify the triple-shape effect. The resulting parts were subjected to resistive heating by passing electric current at different voltages. The printed demonstrators were programmed by a thermo-mechanical programming procedure and the triple-shape effect was realized by increasing the voltage in a stepwise fashion. The 3D printing of such electroactive composites paves the way for more complex shapes with defined geometries and novel methods for triggering shape memory, with potential applications in space, robotics, and actuation technologies.

## 1. Introduction

Additive manufacturing, also known as three-dimensional printing (3DP), has become a topic of considerable interest in recent years due to its ability to realize complex structures at high resolution, which allows design flexibility and prototyping freedom [1,2,3]. The 3DP of smart materials with dynamically tunable shapes coupled with time as the fourth dimension is now referred to as “4D printing” (4DP). Within the family of smart materials, shape-memory polymers (SMPs), with their capability to change shapes upon exposure to various external stimuli such as heat, light, ultrasound, magnetic fields, or chemical substances, are the most investigated 4D-printed materials [4,5,6,7,8,9].

Thermo-sensitive SMPs (trSMPs), with their tailorable elastic properties and transition temperatures, have immense potential in the aerospace, biomedical, electronic and textile industries [10,11,12,13]. Most trSMPs reported to date display a dual-shape effect, changing from one shape to a second shape. At the molecular level, these polymers have crystallites or oriented polymeric chains that act as shape-switching domains associated with a transition temperature. In addition, these polymers contain physical or chemical crosslinks that are responsible for the stability of the permanent shape [14,15]. By introducing multiple types of switching domains with different transition temperatures (*T*_trans_) into one polymer system, it is possible to observe a triple- or a multi-shape effect [16,17,18,19,20,21,22]. The first triple-shape effect was reported in multiphase polymer networks with two types of switching domains, either both crystalline or one crystalline and one amorphous [23]. Furthermore, by using polymer systems with a broad transition temperature or multiple transition temperatures, it was possible to enable a quadruple or quintuple shape-memory effect (SME) [24,25]. Additionally, incorporating functional fillers such as magnetic nanoparticles (MNPs) into triple-shape polymer (TSP) networks, a magnetically controlled triple-shape effect (TSE) was reported [26,27]. These magnetic nanocomposites could be inductively heated by exposure to an alternating magnetic field (AMF), thus enabling a remote-control triple-shape effect. Nevertheless, the processing of network architectures with covalent crosslinks is challenging, limiting the technical use of these composites. Furthermore, the requirement of a power generator and an inductive coil further limits the application potential of magnetically triggered systems. One alternative to inductive heating is Joule heating of electrically conductive composites by passing electric currents, which offers significant advantages over other remote heating methods, such as easy operation, long-range control, and fast response [28]. Carbon-based fillers such as carbon powder, carbon nanotubes, or carbon fibers are incorporated into SMP matrices to enable electric heating. Depending on the type, concentration and level of dispersion, these fillers have the capacity to change the mechanical properties of the composites and provide electrically induced remote heating capabilities [19]. The applicability of electric heating has been investigated for various high-tech applications such as morphing aircraft [29,30], self-deploying structures [31], and intelligent textiles [32]. However, most of these studies have focused on dual-shape effect systems, where the SMP composites were electrically triggered to transition from one shape to another [28]. Electric actuation of TSPs is rare, with few reports published on this topic [19,33]. Nevertheless, control of the two shapes was not possible, and sequential recovery of the two shapes was carried out by the application of a single voltage.

Various 3DP techniques have been used to print single or multicomponent polymer systems to produce 4D objects with enhanced properties and shape-memory capabilities [4,34,35]. For instance, vat photopolymerization (VPP) 3D printing was used to print a photocurable resin, enabling a thermally initiated TSE [34]. Material jetting 3DP has also been used to print a mixture of commercially available photosensitive resins resulting in a TSE [36]. Among different 3DP techniques, material extrusion (MEX) is the most commonly used technique to print 4D objects due to its simple operation and troubleshooting, low cost of equipment and raw materials, high speed, and the capability to print large parts [37,38]. Here, we have explored whether filament-based MEX (i.e., Fused Filament Fabrication (FFF)) can print TSPs, enabling an electrically triggered TSE.

Many thermoplastic SMPs have low stiffness, leading to filament buckling during the FFF printing process. Furthermore, in the case of reinforced composites, filler aggregation can block the nozzle; in both cases, the printing process stops. The issue of filament buckling can be avoided by adding fillers or blending with other polymers to improve filament stiffness [4]. Nevertheless, an optimal amount of filler is required to achieve 3DP by FFF and electrically triggered TSE.

We hypothesized that 4D-printable triple-shape electroactive polymers could be developed by preparing a composite with electric conductivity, multiple switching domains and elasticity. The concept pursued was the creation of an optimal balance between different domains and conductive filler to enable a suitable rigidity for FFF printing and electrically activated TSE. Our strategy involves the fabrication of a multiphase composite by incorporating electrically conductive nanoparticles into a polymer blend with heterogenous morphology containing two switching domains with separate *T*_trans_. The blending of a commercially available thermoplastic polyester urethane (PEU) with poly(lactic acid) (PLA) in the presence of multi-wall carbon nanotubes (MWCNTs) was carried out to fabricate such composites. PLA is an ideal FFF material, which enables a thermally induced SME [39,40]. However, incorporating nanofillers makes PLA brittle, thus limiting its deformability [41]. Therefore, blending polyurethanes (PUs) with PLA was investigated as an effective way to obtain multiphase SMPs with improved strength and elasticity. The PEU selected was a phase-segregated PEU consisting of a crystallizable soft phase based on poly(1,4-butylene adipate) (PBA) and a 4,4′-methylenediphenyl diisocyanate (MDI)/1,4-butanediol (BD)-based hard segment [38,42,43]. The morphology of the composites was explored by using scanning electron microscopy (SEM), transmission electron microscopy (TEM) and atomic force microscopy (AFM). Differential scanning calorimetry (DSC) reveals the separate *T*_trans_ of both switching domains, further confirmed by dynamic mechanical analysis (DMA). Composite filaments of the targeted diameter were easily extruded as monofilaments via screw-based extrusion. Finally, the monofilament was processed in a commercial FFF 3D printer, and smart objects with electrically triggered TSE were fabricated.

## 2. Materials and Methods

A polyesterurethane (PEU) with the tradename Desmopan DP 2795A known for its shape memory capabilities, was received from Covestro Deutschland AG (Leverkusen, Germany). Poly(lactic acid) (PLA) with the tradename Ingeo 4032D was supplied by Nature Works LLC (Plymouth, MN, USA). The PEU and PLA pellets were dried in a vacuum oven at 60 °C overnight before melt processing. Multi-walled carbon nanotubes (MWCNTs) (Graphistrength^®^ C100, diameter: 10–15 nm, length: 1–10 nm) [44] were procured from Arkema (Colombes, France).

Differential scanning calorimetry (DSC) experiments were performed with a Mettler Toledo DSC 3+ (Greifensee, Switzerland), using a heat-cool-heat cycle with constant heating and cooling rates of 2 K·min^−1^ under a nitrogen atmosphere. The sample granulates (7–10 mg) were loaded in Netzsch DSC aluminum pans and sealed. The temperature ranges for the 1st and 2nd heating runs were from 25 °C to 200 °C and −80 °C to 200 °C, respectively. Data from the second heating and first cooling run were used.

Dynamic mechanical analysis (DMA) in tensile mode was carried out on Netzsch Gabo Eplexor 500 N DMA (Ahlden, Germany) equipped with a 25 N load cell using press molded samples with standard dimensions (ISO 527-2/1BB). The measurements were performed in temperature-sweep mode from −100 to 150 °C with a constant heating rate of 2 K·min^−1^ in air, using an oscillation frequency of 10 Hz. During the measurements, a static strain of 1% and a dynamic strain of 0.25% were used. The glass transition (*T*_g_) was determined as the temperature at the maximum in the peak of the loss factor (tan δ) vs. temperature curve.

Scanning electron microscopy (SEM) experiments were performed using a Zeiss Supra 40VP SEM (Carl Zeiss Microscopy Deutschland GmbH, Oberkochen, Germany). For this purpose, planar block faces were prepared in an EMUC6FC6 cryo-ultramicrotome (Leica Microsystems GmbH, Wetzlar, Germany) using a diamond knife at a cutting temperature of −120 °C. Block faces were coated with 5 nm gold in a Q150 R ES sputter coater (Quorum Technologies Ltd., Laughton, UK) and imaged in a high vacuum with an accelerating voltage of 3 kV using an Everhart-Thornley backscattered electron detector. Images were obtained at 2500× to 10,000× magnification.

Transmission Electron Microscopy (TEM) was carried out to see the distribution of MWCNTs in the composite materials. For this purpose, thin films were prepared in an EMUC6FC6 cryo-ultramicrotome (Leica Microsystems GmbH, Wetzlar, Germany) using a diamond knife at a cutting temperature of −120 °C. Sections with thicknesses of 100 to 200 nm were deposited on TEM Grids (Cu, 400 mesh) and examined in a Talos™ F200X TEM (FEI Deutschland GmbH/Thermo Fisher Scientific, Dreieich, Germany) using a Gatan Cryo Transfer Holder Model 914 (AMETEK GmbH, Unterschleissheim, Germany) under cryogenic conditions (−176 °C) at an accelerating voltage of 200 kV in bright field mode. Images were acquired using a Ceta 16M CMOS camera at magnifications of 5000× to 95,000×.

Atomic force microscopy (AFM) was used to determine the phase-specific localization of MWCNTs in the PEU-PLA composites. 2 mm thick sections were cut using a razor blade from the composite and trimmed using a LEICA EM UC6 cryo-ultramicrotome. The microtomed samples were investigated using MFP-3D Infinity (Oxford Instruments, Abingdon, UK) atomic force microscope to obtain the topography and phase contrast images. All the measurements were performed at room temperature, and a standard cantilever holder was used for operation in an air atmosphere. Images were taken with a resolution of 512 × 512 pixels at a 1.5 Hz scan rate. The analysis was performed in Amplitude Modulation-Frequency Modulation (AM−FM) mode using a silicon cantilever AC160TS (Oxford instruments) with a spring constant of about 20–30 N/m. The topography and phase-contrast images were measured at the fundamental resonance frequency of the cantilever (~300 kHz). The images were processed using the Mountains^®^ 9 software (Digital Surf, Besancon, France) to understand the phase specific morphology of the different polymers in the bulk composite. All the measurements were acquired using the same cantilever for a single day.

Wide angle X-ray scattering (WAXS) measurements were conducted at ambient temperature and 55 °C utilizing a Bruker AXS D8 Discover x-ray diffractometer operating in transmission geometry with a two-dimensional HI-Star detector (Bruker, Karlsruhe, Germany). The samples of dimensions 2 × 0.5 cm and thickness 150 μm were fixed at both ends during characterization. The sample-detector distance was set at 150 mm, and the source wavelength was λ = 0.154 nm (Cu K_α_). A graphite monochromator and a pinhole collimator with an opening of 0.8 mm provided a parallel, monochromatic X-ray beam. The two-dimensional diffraction images were integrated to obtain plots with intensity versus diffraction angle (2θ = 5–45°). Both Bragg diffraction peaks from the crystalline phase and the broad scattering peak from the amorphous phase were fitted with Pearson VII functions. The crystallinity index (*X_c_*) was calculated based on the sum of the areas of the fitted peaks assigned to the crystalline phase (*A_cryst_*) and the amorphous phase (*A_amorph_*) (Equation (1)).
(1)$$X_{c}=\frac{A_{cryst}}{A_{cryst}+A_{amorp}}\times 100$$

A custom-built heating device was used to carry out the WAXS measurements of the samples at 55 °C. The sample was equilibrated at this temperature for 5 min before the measurement.

Electric heating and conductivity measurements of the printed composites were analyzed using a Series 2410 SourceMeter (Keithley, Cleveland, OH, USA) with a source voltage range between 5 µV and 100 V and a current range from 10 pA to 1.055 A. Alligator clips were used to connect the printed samples with the source meter. In contrast, the surface temperature was monitored using a VarioCAM^®^ HiRes 384 infrared (IR) camera (InfraTec GmbH, Dresden, Germany).

Composites consisting of a PEU/PLA matrix and MWCNTs as conductive fillers were prepared in a DSM Xplore MC15 HT Vari-Batch micro compounder (Sittard, The Netherlands) in co-rotating mode with a mixing chamber volume of 15 cm^3^. Dried pellets of PEU and PLA were mixed in solid state at a ratio of 70:30 wt%. The content of MWCNT was varied from 5 to 20 wt%. All ingredients were pre-mixed in an aluminum weighing dish via stirring with a spatula, then fed manually into the mixing chamber preheated to 200 °C. Once all the materials were introduced into the mixing chamber, the rotational screw speed was gradually increased from 20 to 100 rpm. The material was recirculated in the mixing chamber for 5 min at 100 rpm to ensure proper mixing. The prepared composites with MWCNT contents from 5 to 20 wt% were tested for their electric heating capacity, as described above. Once it was observed that the addition of 15–20 wt% of MWCNTs was sufficient to realize Joule heating, larger batches of composites containing such levels of MWCNTs were prepared in a Thermo Scientific HAAKE Rheomex PTW16/25 OS co-rotating twin screw compounder (Karlsruhe, Germany). Barrel zone temperatures varied from 190 °C at the hopper to 220 °C at the die, while the screw speed was set to 35 rpm. The dried PEU and PLA pellets and the MWCNTs were pre-mixed manually in a glass container and fed into the hopper of the compounder slowly to avoid exceeding the torque limit. Since it was impossible to ensure the even loading of the ingredients due to granular convection during the manual feeding, it was decided to extrude the compound twice to increase the homogeneity of the mixture. After the first compounding cycle, the extrudate was pelletized using a Thermo Scientific 16 mm fixed-length strand pelletizer (Karlsruhe, Germany). After the second compounding cycle, filaments of a 2.85 mm diameter were extruded and pulled with a Schulz & Busch K-25 conveyor belt (Wülfrath, Germany) in preparation for 3D printing trials.

3D printing was performed in a 3ntr A4v3 filament-based material extrusion machine (Jdeal-Form s.r.l., Oleggio, Italy), also known as a fused filament fabrication (FFF) 3D printer. Computer-aided design (CAD) was performed using the online platform Tinkercad (Autodesk Inc, San Rafael, CA, USA) and the Gcode was prepared using Ultimaker Cura 4.10.0 (Utrecht, The Netherlands). The used FFF 3D printer has three extrusion heads.

Extrusion head 1 uses filaments with a diameter of 1.75 mm, and the other two extrusion heads support filaments with a diameter of 2.85 mm. Four types of specimens were printed with electro-active filaments (i.e., PEU70PLA30MWCNT14). The four printed demonstrators, and their dimensions are shown in Figure 1. The first object was a rectangular bar (50 × 10 × 2 mm^3^) (Figure 1a), which was used for mechanical characterization and quantification of the TSE by bending experiments. A U-shape resistor (Figure 1b) and linear compression device (Figure 1c) were printed with a single electro-active composite. The fourth object (Figure 1d) was a multi-material hinge and was printed with red PLA (passive component) in extrusion head 1 and the composite (electro-active component) in extrusion head 2. Extrusion head 1 had a 0.4 mm brass nozzle, and extrusion head 2 had a 0.8 mm brass nozzle. The printing conditions used to print the different materials are shown in Appendix A).

The triple-shape effect was quantified by measuring the recovery ratios in a triple-shape bending procedure. The 3D-printed composite bar was programmed by a two-step bending procedure. In step one, the sample (shape C) was heated to 90 °C and was bent to 90° (*θ*_B_) and was fixed by cooling to 55 °C under constrain (shape B). After a waiting period of 15 min, the constraint was removed and a subsequent bending to 180° (*θ*_A_) at 55 °C, followed by cooling to −10 °C (shape A), was carried out. After the removal of the constraint, the triple-shape fixation was completed.

For recovery, a stepwise reheating to *T*_mid_ = 55 °C by applying a low voltage (*V*_low_) and to *T*_high_ = 90 °C by application of a higher voltage (*V*_high_) was carried out. The recovery process was recorded via a video camera, and the recovery angles were recorded and evaluated using the ImageJ v1.53e software package (NIH, Bethesda, MD, USA). The ratios of different angles before and after recovery were used to calculate the shape recovery ratios *R*_A–B_ (shape A to shape B) or *R*_A-C_ (shape A to shape C) using Equations (2) and (3). Here, *θ*_B_^rec^ is the angle in the partially recovered sample in shape B, and *θ*_C_^rec^ is the angle in the fully recovered sample (shape C). A schematic demonstration of different angles and shapes during triple-shape bending is shown in Figure 2.
(2)$$R_{A-B}=\frac{\theta_{A}-\theta_{B}\mathrm{rec}}{\theta_{A}-\theta_{B}}\times 100\%$$
(3)$$R_{A-C}=\frac{\theta_{A}-\theta_{C}\mathrm{rec}}{\theta_{A}-\theta_{C}}\times 100\%$$

## 3. Results and Discussions

### 3.1. Selection of Composite Formulation

To retain the flexibility of the blend, PEU was selected as a continuous phase, and the mass ratio of PEU to PLA was varied between 90/10, 80/20, and 70/30. A morphological analysis of these blends confirmed the incompatibility of the PEU and PLA phases. In particular, a two-phase morphology with PLA droplets suspended in the PEU phase can be seen in all blends. Representative SEM images of the pure blends with a PEU-PLA mass ratio of 90/10 (PEU90PLA10) and 80/20 (PEU80PLA20) are shown in Appendix A). The diameter of the PLA domains in the blends was observed to be of the order of ~1–10 microns, with less variability in the case of the PEU70PLA30 formulation. To enable a TSE, the crystallizable soft segments of PEU with a melting point of *T*_m_ = 47 °C served as the first switching domain, while the amorphous segments of PLA with a glass transition of *T*_g_ = 61 °C acted as the second switching domain. The DSC thermograms of neat PEU and PLA are shown in the Appendix A. For the blend PEU/PLA (90/10), no *T*_g_ associated with amorphous PLA was observed, and only a melting transition related to the crystallizable soft segment of PEU was observed. For blends PEU80PLA20 and PEU70PLA30, two different transition temperatures were observed, related to the *T*_m_ of crystallizable segments of the PEU and the *T*_g_ of the amorphous PLA domains, respectively. An investigation of the mechanical properties of the blends by uniaxial tensile testing revealed a significant increase in stiffness (elastic modulus) as the PLA content was increased. The elastic modulus of pure PEU (*E* = 50 MPa) was increased to *E* = 87 MPa for PEU70PLA30. In parallel, the elongation at break (*ε*_b_) was significantly decreased from *ε*_b_ = 860 ± 45% for pure PEU to *ε*_b_ = 319 ± 45% PEU70PLA30. The representative stress–strain curves are shown in Appendix A). To assess the 3D printability of the neat blends, filaments with a uniform diameter d = 2.85 ± 0.05 mm were extruded. One of the major concerns of 3D printing of polyurethanes-based filaments is their lack of stiffness, which can result in buckling in gear-fed 3D printing equipment due to the force applied during the feeding process and the requirement that the filament takes on the role of a piston that applies pressure on the polymer melt. Only PEU70PLA30 (with an elastic modulus of 87 MPa) provided sufficient stiffness to enable FFF printing using the equipment described. For the blends PEU90PLA10 and PEU80PLA20, in contrast, the buckling of the filaments made printing impossible. Based on these initial studies, PEU70PLA30 was chosen for the fabrication of the electro-active composite, given its attractive combination of suitable stiffness for 3D printing and two different transition temperatures for exploring TSE.

MWCNTs were used as filler to increase the electrical conductivity of the polymeric blend, thus allowing for the production of an electroactive TSE. The weight content of the MWCNT in the blends varied between 6 and 18 wt.% (3.3 and 10.5 vol.%). The nomenclature of the composite is given as PEU70PLA30MWCNTx, where “x” is the wt% of MWCNT in the composite as determined by thermogravimetric analysis (TGA) (see Appendix A). Appendix A) shows the electrical resistance of PEU70PLA30 composites with various MWCNT contents at room temperature. The same composite dimensions and testing methodology were used for all formulations, as shown schematically in Appendix A). At filler concentrations up to 11 wt.% (6.3 vol.%), the high resistance of the composite (~1 MΩ) indicates the presence of a polymeric insulating phase. Between 11 wt.% (6.3 vol.%) and 14 wt.% (8 vol.%), a drastic decrease in the resistance to 85 ± 10 Ω was observed. This decrease in resistance was attributed to the formation of conductive pathways in the composite due to exceeding the percolation threshold. By further increasing the content of MWCNTs to 18 wt.% (10.5 vol.%), the resistance was further decreased to 45 ± 10 Ω. Compared to previous reports with a similar type of filler, the percolation threshold observed here was relatively high and implies incomplete dispersion of the MWCNT in the blend [44]. Nevertheless, as this effort focused on achieving electrical conductivity, not filler dispersion, the composites PEU70PLA30MWCNT14 and PEU70PLA30MWCNT18 were selected for further studies, given a low enough resistance to enable Joule heating. In contrast, composites with 11 wt.% of MWCNTs or lower were insufficiently conductive, and no electric heating of these composites was observed. Along with the filler content, the heating efficiency of these electroactive composites depends on the size of the sample, applied voltage, and the time for electric current exposure [45]. Therefore, to assess the heating efficiency and the maximum achievable temperature (*T*_max_), composite specimens with dimensions of 50 × 10 × 2 mm^3^ were printed based on PEU70PLA30MWCNT14 and PEU70PLA30MWCNT18. The current flow and the *T*_max_ achieved due to resistive heating in the composite specimens are shown as a function of voltage in Figure 3. A voltage of 17 V enabled a current flow of 135 mA and a *T*_max_ of 90 °C (well above the PLA *T_g_*) in PEU70PLA30MWCNT14, whereas for PEU70PLA30MWCNT18, only 11 V was required to reach a similar level. Nevertheless, because of the brittle nature of the PEU70PLA30MWCNT18 compound, the PEU70PLA30MWCNT14 was selected for all further investigations of electrically activated TSE.

### 3.2. Morphology of the Composites

The morphologies of the composites on the micro- and nanoscale were elucidated by microscopic analysis. The backscattered SEM images of 3D-printed composite cross-sections of the sample showed irregularly shaped micron-sized aggregates of MWCNT (confirmed by EDX) that were statistically distributed within the polymer matrix. The representative SEM images of PEU70PLA30MWCNT14 are shown in Figure 4a,b. As confirmed by threshold values of MWCNTs, these aggregates enabled the formation of a conductive network in the composite that seems to originate from the high packing density of the primary agglomerates of MWCNTs. In general, due to the high concentration of MWCNTS, strong van der Waals attractions between the individual MWCNTs and poor polymer-MWCNT compatibility, the homogeneous dispersion of MWCNTs in the polymer matrix was not observed. Strategies for achieving higher levels of MWCNT dispersion would include modifying their surface and/or adding compatibilizers to enhance thermodynamic compatibility while reducing their concentration and/or increasing the level of applied shear during mixing to accelerate the kinetics of dispersion. In practice, however, achieving high levels of MWCNT dispersion is not essential for realizing Joule heating. On the contrary, a system with only well-dispersed MWCNTs would contain no MWCNT-MWCNT contacts, precluding the formation of conductive paths and leading to low levels of electrical conductivity, thus making Joule heating challenging to achieve, especially at low voltages [46]. Randomly distributed micro-level voids of different dimensions were also observed in the composite, which could be attributed to the mechanical entrapment of air or the volatilization of small molecules (e.g., water) during melt processing. Furthermore, polymer degradation (either thermal due to shear heating, or hydrolytic due to the presence of water) could also contribute to microvoid formation. In contrast to the observation of these voids, the spherical PLA domains seen in the unfilled blends were not seen in the backscattered SEM images of the composites. This observation is consistent with prior work showing that nanoparticles can cause reductions in the length scale of phase separation in immiscible blends [47].

To further study the distribution of MWCNTs in the polymer blend matrix, TEM analysis was carried out. Here, the difference in electron density resulted in the observation of dark cylindrical features (assigned to MWCNTs) in a lighter matrix (assigned to the PEU/PLA blend), as shown in Figure 4c,d. While individual MWCNTs were observed via TEM imaging and their average diameter was assessed as being ~5–10 nm, no attempt was made to estimate MWCNT length given the very low probability of observing fully intact MWCNTs within a single TEM slice.

The morphology of the 3D-printed composites was further elucidated by AFM analysis. To determine the phase-specific localization of MWCNTs by AFM, four samples were analyzed: neat PEU, a PEU/MWCNT composite (PEU100MWCNT14), the neat PEU-PLA blend (PEU70PLA30), and the PEU-PLA blend/MWCNT composite (PEU70PLA30MWCNT14). AFM images of the composite PEU70PLA30MWCNT14 with sizes 3 × 3 µm^2^ and 1.5 × 1.5 µm^2^ are shown in Figure 5, while the images of neat PEU, PEU100MWCNT14, and PEU70PLA30 are shown in Appendix A). Furthermore, details about the micro-phase separation in pure PEU, the multi-domain architecture of pure blend PEU70PLA30 and the distribution of MWCNT in pure PEU are discussed in Appendix A. In Figure 5, images a–b represent topography, while images c–d represent phase contrast. The blend composite PEU70PLA30MWCNT14 showed a distinct two-phase morphology of PLA and PEU. The topographic images shown in Figure 5a,b reveal a distribution of MWCNTs similar to that observed in Appendix A) for PEU100MWCNT14. Similarly, the relevant phase contrast image (Figure 5c,d) shows a similar level of phase separation between the PLA and the PEU domains, as was observed in Appendix A, for the equivalent MWCNT-free specimen. Furthermore, the phase contrast images of the composite blend indicate selective localization of MWCNTs in the PEU phase. This selective localization phenomenon of MWCNTs in an immiscible blend of thermoplastic polyurethane and PLA was also reported by Buys et al. [48]. Their investigations indicated that whenever inorganic nanoparticles are added to immiscible polymer blends, they tend to be dispersed heterogeneously, either preferentially concentrated in one of the polymer phases or localized to the interface between the two.

### 3.3. Thermal and Thermo-Mechanical Properties

DSC and DMA at varied temperatures were carried out to determine the transition temperatures necessary for the triple-shape programming and recovery process. The DSC thermograms of the neat PEU70PLA30 blend showed a sharp melting transition at *T*_m_ = 49 °C with a melting enthalpy of Δ*H*_m,PEU_ = 34 J·g^−1^ and a weak glass transition just above the terminus of the melting transition (*T*_g_~61 °C) (Figure 6a). The first switching domain was assigned to the melting of the crystallizable PBA soft segments of the PEU, while the second switching domain was associated with the glass transition of PLA. Furthermore, a peak at *T*_c_~10 °C (Δ*H*_c,PEU_ = 28.9 J·g^−1^), corresponding to the crystallization of the PBA soft segments of the PEU, was also observed. By the addition of MWCNTs to the blend to form the composite PEU70PLA30MWCNT14, the *T*_m_ of the PEU soft segments was increased to 50 °C (Δ*H*_m_ = 26.2 J·g^−1^, while the *T*_c_ was decreased to 9 °C (Δ*H*_c_ = 27.5 J·g^−1^) (Figure 6a). However, no significant change in the *T*_g_ of the amorphous PLA domains was observed. Finally, the broad cold crystallization peak (*T*_cc_ = 120 °C) and small melting peak (*T*_m_ = 152 °C, Δ*H*_m,PLA_ = 25 J·g^−1^) assigned to crystalline PLA in the unfilled blend were observed. By adding MWCNTs, Δ*H*_m,PLA_ was slightly increased to 26.2 J·g^−1^, but no significant change in the *T*_m_ of PLA in the composite was observed.

The effects of MWCNT addition on the crystalline microstructure of the PEU-PLA blend composite were further studied by WAXS (Appendix A). For the neat blend PEU70PLA30, the diffraction pattern exhibited clear peaks at 2θ = 21.7° and 24.2° and a shoulder at 22.5°, yielding *χ*_c_ = 18.2 ± 0.3%. The peaks were assigned to the PBA crystalline polymorphs (α form and β form) as indicated in the literature [49,50]. The fact that the characteristic peaks associated with crystalline PLA did not appear may be attributed to the low amount of PLA present in this system. All peaks associated with crystalline polymer disappeared at 55 °C, as expected, given that this exceeds the melting point of PBA as observed via DSC (*T*_m_ = 49 °C). For the composite PEU70PLA30MWCNT14, the addition of MWCNTs resulted in no significant changes in the position and shape of the polymer peaks. For the unfilled blend, no peaks associated with crystalline PLA were observed. The most noticeable change was a characteristic peak associated with the MWCNTs at 2θ = 26° [51]. While, as before, all crystalline polymer peaks disappeared at 55 °C, the MWCNT peak remained, as expected.

Moving from microstructure to viscoelastic response to assess the effects of MWCNTs on the thermomechanical properties of the blend, DMA was carried out as a function of temperature. A stepwise drop in the value of *E*′ was observed in both the unfilled blend and composite as the temperature increased from −50 °C to 100 °C (Figure 6b). *E*′ for PEU70PLA30 decreased from ~1300 MPa at −50 °C to a near-plateau of ~140 MPa at 50 °C. This drop was attributed to the glass transition temperature (*T*_g_) and the melting point (*T*_m_) of the semi-crystalline PBA soft segments of the PEU. A second steep decline in *E*′ was observed between 50 °C and 100 °C, at which point a value of 4 MPa was reached; this was attributed to the *T*_g_ of PLA. Finally, the value of *E*′ started to increase once more at temperatures around 100 °C, the result of the cold crystallization of the PLA [52].

The decrease in modulus at temperatures around 150 °C indicates further softening of the sample before the melting transition detected for the crystalline PLA phase following cold crystallization. However, the amount of PLA crystallinity is too low to explain the large change in modulus fully. It is therefore posited that, in addition to this transition, the MDI/1,4-BDO hard segments present in the PEU are also undergoing a thermal transition. This hypothesis is supported by literature reports reporting hard segment thermal transitions over a similar temperature range [53] and showing that they become difficult to detect via DSC as the hard segment content decreases [54]. Here, given the low modulus of the pure PEU (implying a low hard segment content) coupled with its further dilution by PLA, it is not surprising that the melting transition for the PEU hard segments is not resolved by DSC but remains detectable via DMA. Finally, for the composite PEU70PLA30MWCNT14 formulation (red line in Figure 6b), a higher value of *E*′ was observed across the entire temperature range tested, indicating an increase in rigidity in both the glassy and rubbery regions vs. the unfilled blend.

Uniaxial tensile testing of the neat blend and the composite PEU70PLA30MWCNT14 (50 × 10 × 2 mm^3^) at room temperature and 90 °C was carried out (Figure 6c) to assess quasi-static mechanical properties and deformation capabilities. The elastic modulus of the neat PEU70PLA30 blend was measured as *E* = 87 ± 5 MPa at room temperature, increasing to *E* = 110 ± 3 MPa (+26%) in the case of the filled PEU70PLA30MWCNT14 composition. Nevertheless, the elongation (*ε*_b_) at break was decreased from 319 ± 45% for PEU70PLA30 to *ε*_b_ = 196 ± 15% for PEU70PLA30MWCNT14 (−39%). At an elevated temperature (*T* = 90 °C), the neat PEU70PLA30 blend displayed an elastic modulus of 0.04 MPa and an elongation at break of 140 ± 10%, while the filled PEU70PLA30MWCNT14 formulation gave an elastic modulus of 0.14 MPa (+250%) but could only be extended to 7 ± 2% (−92%) before failing. The limited deformability of the composite could be attributed to the aggregated MWCNTs, which act as stress concentrators and cause premature failure. Due to the low elongation of the printed composites, the electroactive TSE was only explored in bending mode.

### 3.4. 4D Printing of Electro-Active Triple-Shape Composites

The composite PEU70PLA30MWCNT14 exhibited reliable printability and was used to print 3D models of the electroactive composite alone or in combination with electrically passive subcomponents. The printing parameters and the filling density for all the models were kept constant (Appendix A). As explained in the experimental section, the U-shaped composite sample was programmed (*θ*_A_ = 180°) by using a triple-shape bending procedure. To observe the electrically activated TSE, the two ends of the U-shaped specimen were connected using electrodes and subjected to different voltages. An IR camera monitored the surface temperature during electric heating. The application of 13 V enabled a *T*_max_ of 50 °C, above the *T*_m_ of the PBA crystalline domains in the PEU, and triggered a partial recovery (*θ*_B_ = 55°) of the sample in 150 s. Here, it must be clarified that to avoid the involvement of the glass transition range of PLA during the first recovery step, precise heating of the composite to enable the complete melting of the PBA crystalline domain at ~50 °C was necessary. A further increase in voltage level to 17 V resulted in a quick rise of temperature to 90 °C (above the *T*_g_ of PLA) and near-complete recovery of the composite model in 25 s, as shown by thermographic imaging in Figure 7a (Appendix A). The residual angle in shape C can be attributed to the mechanical constraints imposed on the sample by the electrodes connecting to the other ends of the specimen, which restrict complete recovery.

The temperature profile of the composite at two different voltages is shown in Figure 7b. Using the recovery angles during the electrical activation of TSE, the bending recovery ratios $R_{A\to B}$ and $R_{A\to C}$ were calculated. A relatively low recovery ratio $R_{A\to B}$ = 61% for the transition from shape A to shape B was observed. However, a higher value of $R_{A\to C}\;$ = 89% for the second recovery at 17 V was calculated.

The second 3D printed demonstrator is an M-shaped linear compression device, which was printed with an internal angle *θ*_int_ = 20° and was programmed in a triple shape stretching mode. During the triple shape programming step, the internal angle (*θ*_int_) was changed from 15° to 80°. By passing electric current through the device at two different voltages, a stepwise recovery of the device to the original shape was triggered. A voltage of 13 V enabled a change of *θ*_int_ from 80° to 65°, and a further increase of the voltage to 17 V resulted in full recovery of an internal angle of 20° as shown in Figure 8a (Appendix A).

The last 3D-printed object was a multi-material hinge structure. In this object, the hinge itself consists of the electroactive composite with tripe-shape capability, while the outer segments are electro-passive and are based on red-colored PLA. The multi-material 3D structure was thermally programmed using the triple-shape bending procedure described in the experimental section. Electrically induced TSE was triggered by subjecting the composite to different voltages in a stepwise manner. At 13 V, the hinge structure opened partially, increasing the voltage level to 17 V completed the process (Figure 8b). The video of the electrically activated triple-shape capability of the 3D-printed multi-material hinge is shown in the Appendix A.

## 4. Conclusions

In summary, 4D printing of electroactive triple-shape composites based on a PEU/PLA blend filled with MWCNTs was carried out. An optimized selection of the ratios of the three components in the blend enabled smooth FFF printability and Joule heating capability by passing electric current at different voltages. Morphological analysis revealed a phase-separated morphology with randomly distributed agglomerates of MWCNTs in the blend. The melting point of the crystallizable PBA soft segments in the PEU (*T*_m_ = 50 ± 1 °C) acted as the lower transition temperature, while the glass transition of the amorphous PLA (*T*_g_ = 61 ± 1 °C) acted as the upper transition temperature. Different single and multi-material parts were printed using a multi-nozzle FFF 3D printer and programmed using two-step bending procedures. A pronounced TSE was observed by increasing the voltage from 0 V to 13 V and 17 V to achieve well-defined temperatures below, between, and above the PEU and PLA transition temperatures, respectively. The ability to use 4D printing to fabricate composites displaying a stepwise, tunable shape change process that may be triggered remotely on-demand has clear implications for various applications in soft robotics, actuators, and smart textiles.

## Figures and Tables

**Figure 1 polymers-15-00832-f001:**
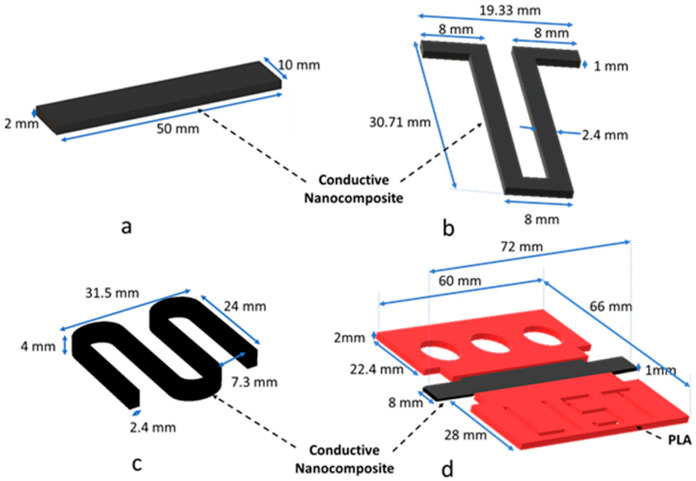
3D-printed specimens with the conductive composite (PEU70PLA30MWCNT14): (**a**) rectangular bar used for mechanical characterization (**b**) U-shaped resistor (**c**) linear compression device; (**d**) hinge 3D printed with non-conductive PLA (red) and a conductive composite (black).

**Figure 2 polymers-15-00832-f002:**
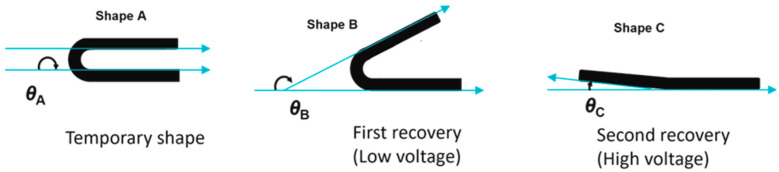
A schematic demonstration of the triple-shape bending procedure.

**Figure 3 polymers-15-00832-f003:**
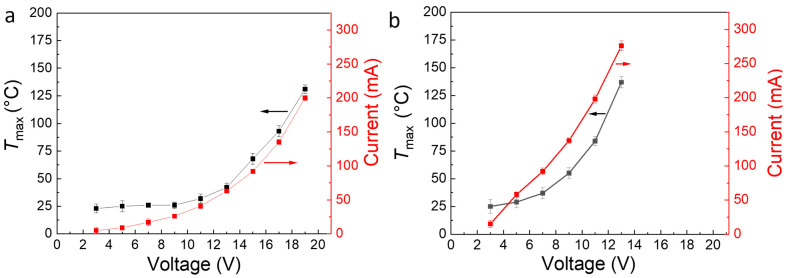
Maximum achievable temperatures (*T*_max_) in composite samples (**a**) PEU70PLA30MWCNT14 and (**b**) PEU70PLA30MWCNT18 by passing current at variable voltages.

**Figure 4 polymers-15-00832-f004:**
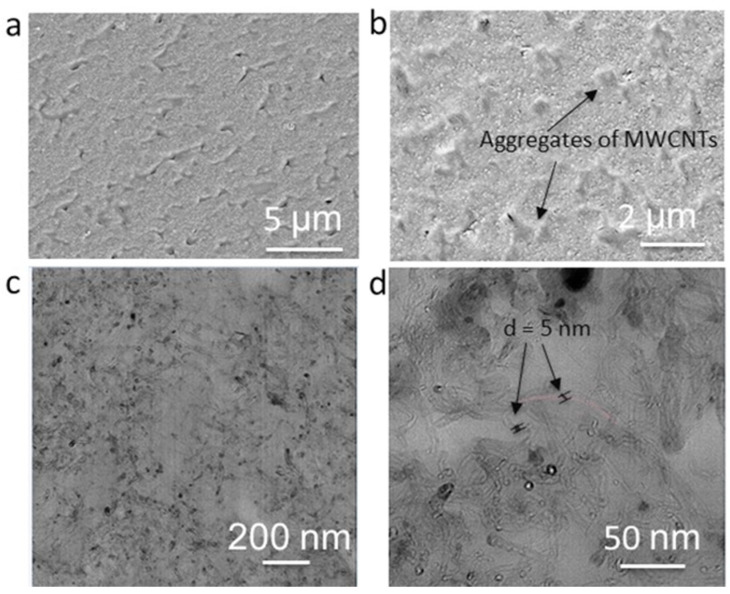
(**a**,**b**) Backscattered SEM images of the PEU70PLA30MWCNT14 and (**c**,**d**) TEM analysis of the composite.

**Figure 5 polymers-15-00832-f005:**
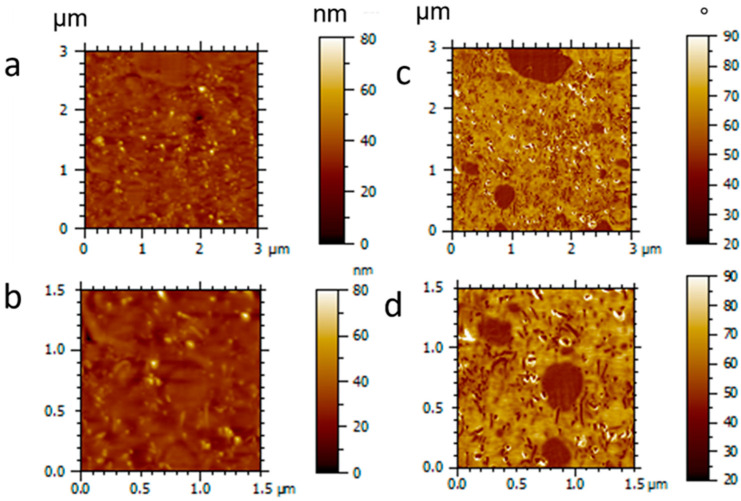
AFM images of the blend composite PEU70PLA30MWCNT14 (sizes 3 × 3 µm^2^ and 1.5 × 1.5 µm^2^; Images (**a**,**b**) are topography images, (**c**,**d**) are phase contrast images.

**Figure 6 polymers-15-00832-f006:**
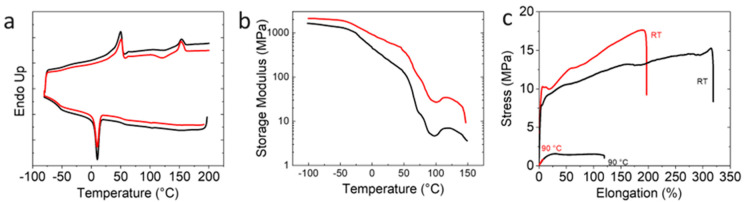
(**a**) DSC thermograms of neat blend and blend composite (**b**) Storage modulus at varied temperatures (**c**) Uniaxial tensile testing of the neat blend polymer and composite at room temperature (RT) and 90 °C. PEU70PLA30 (black line), PEU70PLA30MWCNT14 (red line).

**Figure 7 polymers-15-00832-f007:**
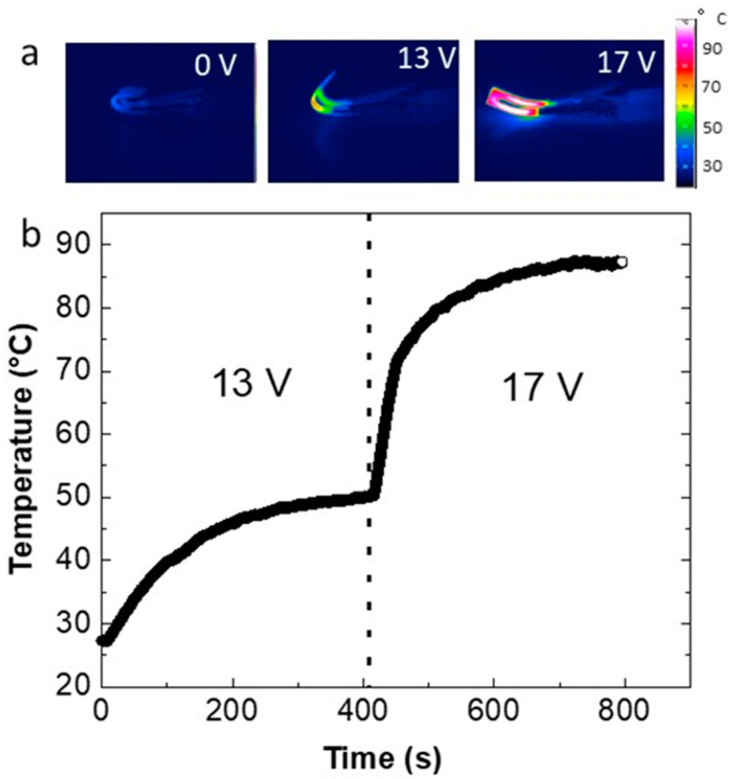
(**a**) IR thermographic images of the electrically triggered TSE of a “U” shaped composite part by a stepwise increase in voltage from 13 V to 17 V. (**b**) The temperature profile of the composite sample when subjected to two different voltages.

**Figure 8 polymers-15-00832-f008:**
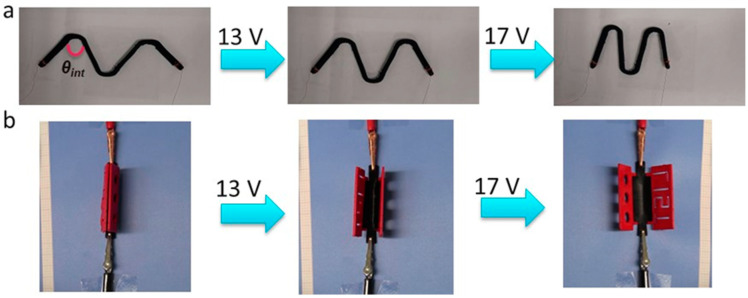
Electrically triggered TSE in 3D-printed composites samples (**a**) “M” shaped sample based the electroactive composite, (**b**) multi-material hinge structure consisting of electroactive composite (black) and passive PLA subcomponents (red).

## Data Availability

Not applicable.

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
