# Peer review of "4D Printing of Electroactive Triple-Shape Composites"

_polymers, 2023, doi:10.3390/polym15040832_

Round 1

Reviewer 1 Report

In this work, the authors report the 4D printing of electro-active triple-shape composites based on thermoplastic blends (PEU/PLA) filed with multiwall carbon nanotubes. The authors performed in-depth morphological, thermal, and thermos-mechanical analyses and subsequently printed 3D demonstrators. These results will surely impact the soft robotics community.

I have some minor comments I would like to see addressed, but overall, I consider it suitable for publication.

  1. In materials and methods, I believe there is a misspelling of Besançon.
  2. The discussion regarding the morphology analysis is interesting and should be kept, however I would recommend moving it (or moving a part of it) into the supplementary file. Line 6 does not seem to belong to the discussion and should be removed and re-phrased.
  3. The quality of the figures needs to be revised.
  4. The title of the part 3.4 contains a misspelling (3.4 4D printing instead of 3.4.4 D printing).

Reviewer 2 Report

In this Manuscript “4D Printing of Electroactive Triple-Shape Composites”, the authors 4D-printed electro-active triple shape composites via fused filament fabrication (FFF) by using composite blends of polyester urethane (PEU), polylactic acid (PLA), and multiwall carbon nanotubes (MWCNTs). Additionally, the authors also evaluated their triple shape memory behavior, which is highly suitable for developing soft robots, actuators or space components.

Overall, the article looks novel and interesting. However, some shortcomings need to be addressed before possible publication in polymers.

Please find the attached annotated file to see my comments.

Based on my comments, the recommendation is Major Revision.

Round 2

Reviewer 2 Report

It can be accepted now